# A Novel Longitudinal Phenotype–Genotype Association Study Based on Deep Feature Extraction and Hypergraph Models for Alzheimer’s Disease

**DOI:** 10.3390/biom13050728

**Published:** 2023-04-23

**Authors:** Wei Kong, Yufang Xu, Shuaiqun Wang, Kai Wei, Gen Wen, Yaling Yu, Yuemin Zhu

**Affiliations:** 1College of Information Engineering, Shanghai Maritime University, 1550 Haigang Ave., Shanghai 201306, China; 2Bio-Med Big Data Center, CAS Key Laboratory of Computational Biology, Shanghai Institute of Nutrition and Health, University of Chinese Academy of Sciences, Chinese Academy of Sciences, Shanghai 200031, China; 3Department of Orthopedic Surgery, Shanghai Jiao Tong University Affiliated Sixth People’s Hospital, Shanghai 200233, China; 4Institute of Microsurgery on Extremities, Shanghai Jiao Tong University Affiliated Sixth People’s Hospital, Shanghai 200233, China; 5CREATIS UMR 5220, U1294, CNRS, Inserm, INSA Lyon, University Lyon, 69621 Lyon, France

**Keywords:** Alzheimer’s disease, brain imaging genetics, gene expression, deep subspace reconstruction, hypergraph, DS-HBTGSCCA

## Abstract

Traditional image genetics primarily uses linear models to investigate the relationship between brain image data and genetic data for Alzheimer’s disease (AD) and does not take into account the dynamic changes in brain phenotype and connectivity data across time between different brain areas. In this work, we proposed a novel method that combined Deep Subspace reconstruction with Hypergraph-Based Temporally-constrained Group Sparse Canonical Correlation Analysis (DS-HBTGSCCA) to discover the deep association between longitudinal phenotypes and genotypes. The proposed method made full use of dynamic high-order correlation between brain regions. In this method, the deep subspace reconstruction technique was applied to retrieve the nonlinear properties of the original data, and hypergraphs were used to mine the high-order correlation between two types of rebuilt data. The molecular biological analysis of the experimental findings demonstrated that our algorithm was capable of extracting more valuable time series correlation from the real data obtained by the AD neuroimaging program and finding AD biomarkers across multiple time points. Additionally, we used regression analysis to verify the close relationship between the extracted top brain areas and top genes and found the deep subspace reconstruction approach with a multi-layer neural network was helpful in enhancing clustering performance.

## 1. Introduction

Alzheimer’s disease (AD) is a progressive neurodegenerative disease and has a devastating financial and emotional impact on individuals and their families [1]. Memory loss, cognitive decline, and behavioral problems are the main clinical signs of AD, which will continue to progress over time [2]. Regional variations in the disease’s progression will occur at various times and in various parts of the brain [3]. Numerous studies have discovered that these regional variations may be influenced by different single nucleotide polymorphisms (SNPs) at different times. The study of how complicated genetics affects temporal patterns in imaging genetics has gained popularity [4,5]. To realize early intervention and treatment for AD and stop AD’s progression, finding biomarkers in the disease’s progression is advantageous [6].

To determine the correlation between imaging genetics data, the majority of early imaging genetics relied on linear models, such as the Sparse Canonical Correlation Analysis (SCCA) algorithm [7]. SCCA is a standard high-dimensional data analysis method based on CCA [8]. By adjusting the weights of different properties of various modal data, it can maximize the correlation of two different modal data. Based on SCCA, Fang et al. incorporated the concept of a joint sparse model and fusion graph lasso. They presented a Joint Sparse Canonical Correlation Analysis (JSCCA) technique to increase the accuracy of the algorithm for the joint analysis of high-dimensional imaging data [9]. Furthermore, Kim et al. extended JSCCA to include connectivity-based penalties, validating the robustness of the algorithm to select potential biomarkers associated with Parkinson’s disease on both simulated and real imaging genetics datasets [10]. The link between SNPs and quantitative traits (QTs) can also be examined using the sparse additive model (SpAM). Compared to traditional linear models, it can produce a more realistic impact when taking into account the nonlinear effect of SNP [11]. However, the majority of these methods focus on the relationship between quantitative traits (QTs) and SNPs from a single time point, which may disregard the wealth of historical data that the evolving QT carries through numerous time points. In order to explore the interaction between QTs and genetic data in brain imaging, several researchers have taken into account the time structure of longitudinal imaging data. The most typical method is to improve the SCCA algorithm by including sparse terms to help in the identification of pertinent markers and the construction of a diagnostic model, enabling the algorithm to locate a few strong correlation pairs in high-dimensional imaging genetics data. A valuable temporally constrained group sparse canonical correlation analysis (TGSCCA) algorithm was proposed by Hao et al. [12] to identify the relationship between SNPs and longitudinal sMRI phenotypes. Du et al. [4] proposed the multi-task sparse canonical correlation analysis (T-MTSCCA) to analyze how SNPs change brain QT over time using longitudinal neuroimaging data. Recent research has begun to take joint multimodal longitudinal regression and classification into account to find AD biomarkers and associations between data from various modalities [13].

Considering that linear models may not be able to reveal how genetic factors affect the structure and function of the brain through complex nonlinear relationships, and most of the current methods for analyzing image genetic data only apply QTs from a single time point (e.g., baseline), in this study, we proposed a novel algorithm named DS-HBTGSCCA that combines the deep subspace reconstruction and hypergraph-based Time Constraint Group SCCA (TGSCCA) algorithms to analyze the bivariate correlation between reconstructed longitudinal structural magnetic resonance imaging (sMRI) data and gene expression data. Firstly, the deep subspace reconstruction algorithm was used to reconstruct the original data, and the unique expression patterns of different label samples were identified. Specifically, the deep subspace reconstruction approach transferred each sample to a nonlinear feature space via a multilayer feedforward neural network to extract the nonlinear link between regions of interest (ROIs) and gene variants. Furthermore, it was innovative to maintain the high-order structural relationship between the identical modal samples by using the hypergraph’s Laplacian matrix and K nearest neighbors (KNN) to embed the hyperedge and treat each feature as a vertex. Finally, we performed a biological significance analysis and regression analysis of top brain regions on the top risk brain regions and top genes mined, and verified the superiority of the reconstructed data.

## 2. Methods

### 2.1. Data Source and Preprocessing

The gene expression data and longitudinal brain imaging data of 210 non-Hispanic white participants were downloaded from the Alzheimer’s Disease Neuroimaging Disease (ADNI) website (https://adni.loni.usc.edu/ (accessed on 15 March 2022)), including 10 AD, 102 Early Mild Cognitive Impairment (EMCI), 36 Late Mild Cognitive Impairment (LMCI), and 62 Healthy Control (HC) subjects. Specifically, all the data we used in this experiment came from the data of the ADNI1 queue. Because the longitudinal image genetics research in this paper needs to collect sMRI and gene expression data of the same subject at multiple time points, the interval between time points needs to be roughly consistent. Therefore, only 10 AD samples meet our screening requirements. The demographics of the participants are listed in Table 1.

#### 2.1.1. Preprocessing of Longitudinal Imaging Data

From the ADNI database, we retrieved the sMRI data for the samples with the time distributions of baseline (T1), 6th month (T2), 12th month (T3), and 24th month (T4). After selecting, we eliminated brain imaging samples presenting illegible structures and samples having distorted images, leaving us with 210 samples in total. The four brain images of each sample then underwent head motion correction segmentation, registration, and feature extraction using the SPM12 program in MATLAB. After extracting 122 ROIs using the Anatomical Automatic Labeling (AAL) template, 32 cerebellar structures were eliminated and 90 ROIs were kept as the final characteristics of the brain image.

#### 2.1.2. Preprocessing of Gene Expression Data

For the original gene expression data downloaded from the ADNI website, differential expression analysis was performed on 210 samples (diseased group: 148; healthy control group: 62) using the limma package of R [14], and genes with *p* < 0.05 were included, and finally 413 significantly differential expression genes were obtained as features of gene expression data.

### 2.2. Deep Subspace Reconstruction Algorithm

Because of the large amount of noise in biological data, the estimation bias will be introduced when the biological data are constrained by a network or graph. Therefore, considering the multi-subspace structure of the bottom data layer, samples with the same label are gathered in the same subspace, and different samples are distributed in different subspaces. The objective function of the deep subspace reconstruction algorithm [15,16] is defined as:
$$\underset{W^{x}}{\min}{\left\|W^{x}\right\|}_{p}$$(1)$$s.t.X=W^{x}X,\left(\mathrm{diag}\left(W^{x}\right)=0\right)$$
where $W^{x}$ is the self-expression coefficient matrix, and X is the feature matrix of the original data.

Data label information is a crucial piece of prior knowledge, and it is possible to find greater correlation between imaging and genetic data by putting tags into the algorithm. On this basis, the following objective function can be obtained by incorporating the tag information:(2)$$\underset{{\left(W_{i}^{x}\right)}^{\left(d\right)}\geq 0}{\min}\frac{1}{2}{\left\|{x_{i}}^{\left(d\right)}-{\left(\tilde{X}\right)}^{\left(d\right)}{\left(W_{i}^{x}\right)}^{\left(d\right)}\right\|}_{2}^{2}+\lambda {\left\|{\left(W_{i}^{x}\right)}^{\left(d\right)}\right\|}_{1}$$
where $\tilde{X}$ represents the sample set of all labels, $x_{i}^{\left(d\right)}$ represents the *i*-th sample in the *d*-th label, and ${\left(W_{i}^{x}\right)}^{\left(d\right)}$ represents the self-expression coefficient matrix of the *i*-th sample in the *d*-th lable.

Nonlinearity, high dimension, and small sample size have always been the limitations of SCCA when trying to calculate the correlation between two variables. Therefore, in our study, the deep subspace reconstruction algorithm was used to reconstruct longitudinal sMRI data and gene expression data. By introducing a multilayer feedforward neural network, each sample’s brain network QTs and gene expression data can be built in a nonlinear space. The schematic diagram of deep subspace reconstruction is depicted in Appendix A.

### 2.3. Hypergraph Learning

Ordinary graphs can be used to describe the binary relationship between two objects, whereas hypergraphs [17] have the ability to connect more than two vertices together through hyperedges to exploit the relationship between multiple related objects. By using the prior knowledge between nodes, one can create hyperedges and, subsequently, hypergraphs. High-order correlations between samples belonging to the same modal can be maintained using the Laplace matrix of hypergraphs.

We set $G\left(V,E,W\right)$, which represents a hypergraph, where V is the set of vertices, E is the set of hyperedges, and W is the set of weights of the hyperedges. Each hyperedge $e_{i}\left(i=1,2,\ldots N_{e}\right)$ is assigned a weight $a(e_{i})$. Appendix A is a hypergraph, in which each hyperedge can connect multiple nodes. For the given hypergraph G, its incidence matrix H is defined as follows:(3)$$H\left(v,e\right)=\left\{\begin{array}{c} 1,\;if\;v\in e\\ 0,if\;v\notin e \end{array}\right.$$

Based on the incidence matrix H, the degree v of each vertex and the degree e of each hyperedge are, respectively, defined as follows:(4)$$d\left(v\right)=\underset{e\in E}{\sum }w\left(e\right)H\left(v,e\right)\;$$
(5)$$\delta \left(e\right)=\underset{v\in V}{\sum }H\left(v,e\right)\;$$

In addition, let $D_{v}$ and $D_{e}$ represent the diagonal matrices corresponding to the degrees of vertices and the degrees of hyperedges, respectively. W represents a diagonal matrix containing hyperedge weights.
(6)$$D_{v}=diag\left(d\left(v_{1}\right),\ldots ,d\left(D_{v_{N}}\right)\right)\;$$
(7)$$D_{e}=diag\left(d\left(e_{1}\right),\ldots ,d\left(D_{e_{N}}\right)\right)\;$$
(8)$$W_{ii}=a(e_{i})\;$$

In this paper, we adopt the method proposed in [18] to compute the hypergraph Laplacian matrix:(9)$$L^{h}=I-\Theta \;$$

Among them, $L^{h}$ is the hypergraph Laplacian matrix, where I is the identity matrix and $\Theta =D_{v}^{-1/2}HWD_{e}^{-1}H^{T}D_{v}^{-1/2}$.

### 2.4. The Proposed Optimization Algorithm

This paper merged the hypergraph matrix of two kinds of omics data after deep subspace reconstruction into the TGSCCA algorithm and induced the new algorithm (DS-HBTGSCCA) to find the high-order relationship between omics data from the sample relationship provided by different omics data (the high-order relationship between longitudinal image data and gene expression data is shown in Appendix A). The system frame diagram is shown in Figure 1.

The original gene expression data $X^{*}$ and longitudinal imaging phenotype data $Y_{t}^{*}$ are reconstructed in depth subspace to obtain ***X*** and $Y_{t}$, where $X^{*}={\left[x_{1},\ldots ,x_{n},\ldots ,x_{N}\right]}^{T}\in R^{N\times p}$ is the gene expression data of N subjects, $Y_{t}^{*}={\left[y_{1t},\ldots ,y_{nt},\ldots ,y_{Nt}\right]}^{T}\in R^{N\times q}$ is longitudinal imaging phenotype data, $L_{h_{1}}$ and $L_{h_{2}}$ are the hypergraph matrices of ***X*** and ***Y***, respectively. Hypergraph constraints can be rewritten with *P* ($H_{i}$) (*i* = 1,2) as follows:(10)$$P\left(H_{1}\right)=T_{r}\left(H_{1}^{T}B_{1}H_{1}\right)\;$$
(11)$$\;P\left(H_{2}\right)=T_{r}\left(H_{2}^{T}B_{2}H_{2}\right)\;$$
$\mathrm{where}\;B_{1}$ and $B_{2}$ are the Laplace matrices of ***X*** and ***Y***, respectively, and $\;B_{1}=L_{h_{1}}$*,* $B_{2}=L_{h_{2}}$.

Based on the TGSCCA model [13], the optimization algorithm function can be written as:(12)$$\underset{u,V}{\min}\overset{T}{\underset{t=1}{\sum }}u^{T}X^{T}Y_{t}v_{t}+\lambda_{1}∥u{∥}_{1}+\lambda_{2}{\left\|V\right\|}_{2,1}+\lambda_{t}\overset{T-1}{\underset{t=1}{\sum }}\|v_{t+1}-v_{t}\|-\beta_{1}T_{r}\left(H_{1}^{T}B_{1}H_{1}\right)-\beta_{2}T_{r}\left(H_{2}^{T}B_{2}H_{2}\right)$$
where *T* is the number of time points and $V=\left[v_{1},\ldots ,v_{t},\ldots ,v_{T}\right]\in R^{q\times T}$.

### 2.5. Performance Index of the Algorithm

In this paper, we take the canonical correlation coefficient (*CCC*) as the performance index of the algorithm. The calculation formula of this index is given below.
(13)$$CCC_{t}=Corr\left(Xu,\;Yv_{t}\right)$$

Among them, $CCC_{t}$ stands for *CCC* of correlation analysis of two kinds of data at the t time. $Corr\left(\cdot \right)$ stands for Pearson *CCC*.

### 2.6. Regression Index

*R^2^* score is recommended to evaluate the benefits and drawbacks of regression models. The formula for calculating *R^2^*_score is:(14)$$R^{2}=1-\frac{SSE}{SST}$$

It can be simplified to:(15)$$R^{2}=1-\frac{\sum_{i=1}^{n}{\left(y_{i}-\hat{y_{i}}\right)}^{2}/n}{\sum_{i=1}^{n}{\left(y_{i}-\bar{y}\right)}^{2}/n}=1-\frac{MSE}{Var}$$
where *MSE* stands for mean squared error and *Var* stands for variance.

### 2.7. Statistical Analysis

Subjects’ age statistics were presented in the form of mean ± standard deviation (SD) and statistically analyzed using the *t*-test of SPSS 20.0 software. Figure 1 was drawn by using online software called ProcessOn (https://www.processon.com (accessed on 30 November 2022)). Figure 2 was created by using Microsoft Excel. Matlab 2018a was used to draw Figure 3. Figure 4 was displayed by using BrainNet Viewer of Matlab. Figure 5 was displayed by Matlab 2018a. Figure 6 and Figure 7 were obtained by using R package “ggplot2”. Adobe illustrator was applied to splice multiple pictures.

## 3. Results

### 3.1. On Simulation Data

To evaluate the performance of the proposed DS-HBTGSCCA, we simulated neuroimaging datasets ${\;T}_{i}\;\left(i=1,2,3,4\right)$ and genetic dataset X with n samples, where $T_{i}$ has p features, and X includes q features. We set n = 200, p = 90, and q = 450. Assuming that the two data are correlated, α represents the sMRI feature weight vector with p elements, and β represents the gene feature weight vector with q elements, where each element independently obeys a uniform distribution $U\left(-1,-0.5\right)\cup U\left(0.5,1\right)$. The correlation between sMRI and genes can be simulated by using a latent variable ε with a normal distribution $N\left(0,\sigma_{\epsilon }^{2}\right)$. For sMRI data, $T_{i}=\beta \epsilon +e$ represents correlated voxels, $T_{i}=e$ represents uncorrelated voxels and e represents noise with the normal distribution $N\left(0,\sigma_{e}^{2}\right)$. $T_{i+1}={\;T}_{i}+\Delta v\;\left(\Delta v\;\sim \;N\;\left(0,\;0.1\right)\right)$ and i=1,2,3. For gene expression data,  X=αε+e and X=e are generated to represent correlated and uncorrelated voxels, respectively. The noise level of the simulation data set was set from 1 to 5, which was employed to compare the performance of the proposed algorithm and the TGSCCA. Then, on the generated 200 simulation data samples, the training set and the test set were divided according to 4:1, and the model was trained on the training set to obtain the optimal hyperparametric solution. The parameter tuning range is {0.0001, 0.001, 0.01, 0.1, 1}. Each loop’s average of the *CCCs* from the four epochs and its used hyperparameters were released at the end of each loop. Next, the trained model from the test set was utilized to calculate the correlation of each period, and the corresponding weight and *CCC* were then calculated. Taking TGSCCA as the control group, the *CCCs* of the two algorithms in four periods under different noise environments were obtained (Appendix A). In addition, we drew corresponding line charts through the data of Appendix A, so as to better observe the changes in the data (Figure 2).

As shown in Figure 2, we evaluated the *CCC* of DS-HBTGSCCA and TGSCCA at different noise levels as a performance indicator. The results showed that the two algorithms obtained the largest *CCC* when noise = 2. In T1, T2, and T3 periods, our algorithm was better than TGSCCA. Only in the T4 period, the *CCC* of TGSCCA was greater than DS-HBTGSCCA, but there were not many differences in the two algorithms. In other noise cases, the *CCC* of DS-HBTGSCCA was larger than that of TGSCCA. This indicated that our approach was more effective at detecting longitudinal joints than TGSCCA and had good robustness.

### 3.2. On Real Imaging Genetic Data

This experiment made use of the real data in Section 2.1. To prove the superiority of the reconstructed data, we conducted experiments on the optimized model using both the original and the reconstructed data. The dividing method and parameter setting of the training set and the test set of the two data were consistent with the simulated data. Table 2 and Table 3 are *CCCs* on the reconstructed and the original data using two methods on ADNI data, respectively. Table 1 shows that the correlation of DS-HBTGSCCA was higher than the TGSCCA algorithm’s correlation in the T1 and T2 periods. The difference between the two in the T3 period was not statistically significant, and our algorithm’s *CCC* was lower than the TGSCCA algorithm’s correlation in the T4 period. Meanwhile, the *CCCs* of the reconstructed data for both algorithms were higher than the *CCCs* of the original data, further demonstrating that the data reconstructed by the deep subspace reconstruction method and hypergraphs can more accurately depict the relationship between longitudinal sMRI data and gene expression data. In a word, the correlation performance of the DS-HBTGSCCA algorithm was better than TGSCCA in real data at most times.

### 3.3. Top ROIs and Top Genes Identification

The heatmaps of the brain’s regions and the gene weight values obtained using the two different methods on reconstructed data are shown in Figure 4. DS-HBTGSCCA can uncover consistent patterns at four time points in the process of examining brain risk regions and risk genes, whereas the TGSCCA algorithm can only be employed at certain time points. This implies that DS-HBTGSCCA improved the correlation performance between longitudinal sMRI data and gene expression data and that our method was highly stable for detecting longitudinal joints.

The top 10 ROIs on the reconstructed longitudinal sMRI, as determined by our method and TGSCCA, are listed in Table 4**,** along with their weights (absolute values). Compared with TGSCCA, DS-HBTGSCCA could identify the brain regions with the same weights in four time points because of the innovative addition of deep subspace reconstruction to reconstruct the original data and the high-order correlation of the reconstructed data by hypergraph mining, which signifies that DS-HBTGSCCA could jointly select brain regions with the same capture time pattern in ROIs from adjacent time points and had strong robustness. The top 20 genes determined by DS-HBTGSCCA are provided in Table 5, together with their canonical weights (absolute values) based on reconstructed gene expression data.

Figure 4 displays a graphic mapping of the top 10 ROIs on the reconstructed longitudinal sMRI. To determine whether there was a connection between nodes, the Pearson CCC (PCC) between the top 10 ROIs was calculated, and the mean value of PCC (0.9798) was calculated. In this figure, the size of the nodes was represented by the weight calculated in accordance with CCA. The two brain regions with values greater than the mean had a connectivity and relationship with one another.

In order to study the influence of time series ROI changes, we performed an independent sample *t*-test between the four periods of T1–T4 for the top 10 brain regions mentioned and found that only two brain regions (left middle temporal (*p* = 0.025) and right inferior temporal (*p* = 0.04)) between T1 and T4 had significant differences. Then, we draw the heatmaps of the two brain regions with significant differences and the top 20 genes for HC and AD in T1 and T4 (Figure 5). Figure 6 shows the enrichment analysis results of the Top 10 genes to explore the relationship between the top 20 differentially expressed genes selected by the DS-HBTGSCCA and see their functions clearly.

### 3.4. Regression Results Using Selected Top Markers

In this section, we utilized gene expression data to forecast the expression of gray matter volume in real data in order to predict the regression response. The smaller the error of response, the closer the relationship between selected genes and gray matter volume, and the stronger the representativeness. Regression analysis was carried out specifically using Bayesian regression, linear regression, and ridge regression. The gray matter volume of the top 10 brain regions was regressed using the top 10 to top 100 genes, respectively. The mean *R^2^*_score obtained from the reconstructed data by the three regression methods is given in Table 6.

### 3.5. Effectiveness Verification Results of Deep Subspace Reconstruction

To evaluate the influence of the deep neural network on subspace clustering performance, we tested the clustering effect of multi-omics data with and without a multilayer neural network and used six different indices (contour coefficient, F-score, Precision, Recall, normalized mutual information (NMI), adjusted rand index (Adj-RI)) to evaluate the clustering performance (Table 7). The higher the value of the indices, the better the clustering effect.

We also evaluated the effectiveness of the algorithm reconstruction by using the K-means algorithm to cluster based on the reconstructed T1, T2, T3, T4, and gene data. Then, we applied t-distributed stochastic neighbor embedding (t-SNE) to reduce the dimension for visualization (Appendix A). We introduced three clustering indicators (silhouette_score, calinski_harabasz_score, and davies_bouldin_score) to evaluate the reconstruction effect (Appendix A). Among them, the bigger the silhouette_score and calinski_harabasz_score, the smaller the davies_bouldin_score.

Based on Table 2 and Table 3, we discovered that the reconstructed data have higher *CCCs* across epochs on our model than the original data. Subsequently, we identified temporally consistent brain areas and genes. Taking T1 period as an example for longitudinal images, we drew heatmaps of the top 10 ROIs and top 20 genes identified by DS-HBTGSCCA on the reconstructed and original data (Figure 7).

## 4. Discussion

### 4.1. Biological Significance Analysis of Top ROIs

MRI can be used to evaluate the pattern of alterations in the gray and white matter of AD brains to advance AD research. Based on the effectiveness of the DS-HBTGSCCA, the top 10 ROIs have mostly been documented to be associated with AD. For example, researchers discovered that the left middle temporal lobes were closely associated to AD in the experiment evaluating the effect of tandospirone citrate on the treatment of AD [19]. Left precuneus, right precuneus, left precentral gyrus, and right postcentral gyrus were identified to be associated to AD by creating a multi-layer network model of NC, MCI, and AD [20]. In MCI patients, the right language network may be damaged, and the right middle temporal gyrus is abnormal [21]. In healthy middle-aged individuals, AD risk genes also change the default mode network’s (DMN) functional connection mode, which results in differences in the left middle frontal lobe [22].

### 4.2. Correlation Analysis between Time-Series Related ROIs and Top Genes

We found a correlation between the top brain regions with time series and top genes. It was clear that more powerful correlation pairs were captured in the transition from HC to AD samples (Figure 5). The middle temporal gyrus is involved in many different brain functions, such as thinking and recognizing faces. The alexia and agraphia of Chinese characters are also caused by injury to the posterior part of the middle temporal gyrus in the left cerebral hemisphere [23]. The inferior temporal gyrus is associated with abnormal semantic memory disorders (such as AD) [24]. The inhibitor of translation initiation of the EIF4EBP2 gene involves synaptic plasticity, which can affect learning and memory formation through similarity. RTN4.1 is a member of the RTN4 family. RTN4-related diseases include temporal lobe epilepsy (TLE), which is a chronic disease of the nervous system, and its related pathways include AD and miRNA effect (https://www.genecards.org/cgi-bin/carddisp.pl?gene=RTN4&keywords=RTN4 (accessed on 16 March 2023)). Research has shown that TLE is related to memory impairment and memory loss. The loss of pyramidal cells in TLE will cause verbal memory defects, and the loss of right neurons is more prominent in non-verbal (visual-spatial memory loss). Therefore, it can be inferred that EIF4EBP2 is related to the inferior temporal gyrus, which can cause memory impairment, and RTN4 is related to the middle temporal gyrus, which can cause alexia. Future research can find out the expression of these two genes in neurodegenerative diseases such as AD and determine their relationship [25,26,27].

### 4.3. Biological Significance Analysis of Top Genes

Many human diseases are intrinsically caused by genes. It is possible to forecast AD, administer the proper pharmacological intervention, halt the progression of AD, and significantly contribute to AD clinical research by researching the association between genes and AD and finding genes connected to AD. All of the aforementioned studies conclusively demonstrated that the top 20 genes (FCER1G, AIF1.1, EIF4EBP2, MORF4L1, PRR13, CCNY.2, TUBB1, MAPKAPK3.1, IER3, PECAM1, CELF2.1, CORO1C.2, RTN4.1, XRCC5), which account for 70% of the disease, were either directly or indirectly associated to the development of AD. FCER1G, which has a strong negative correlation with the right middle temporal of AD in T1 period, is closely related to the pathophysiology of Aβ, and the expression level of FCER1G affects the progress of AD [28]. Inflammation and apoptosis may all lead to AD [29,30,31]. MAPKAPK3.1 and CORO1C.2 were negatively correlated with right middle temporal in AD samples at T1 and T4, in which MAPKAPK3 was involved in regulating the inflammatory response of mammals [32], while CORO1C was related to apoptosis. It can be inferred that MAPKAPK3.1 belonging to mapkapk3 family and CORO1C.2 belonging to coro1c family may also participate in the pathological process of AD. Evidence has also shown that other genes with no obvious positive and negative correlation may play an important role in AD. Sanfilippo et al. [33] found that the expression level of PECAM1 in the brain of AD patients was regulated. PRR13 negatively regulates thrombospondin-1 (TSP1) expression at the level of transcription. This down-regulation was shown to reduce apoptosis. The studies in [34,35] all confirmed that TSP-1 is a potential therapeutic target of AD pathogenesis. Future research needs to examine how the expression of other genes changes throughout time to affect how AD develops.

The results of GO analysis showed that these genes’ pathways are almost entirely related to AD, which demonstrated that our approach had a substantial benefit in terms of feature selection (Figure 6). It was obvious from the figure that these pathways’ biological processes were most intensive when they relate to neutrophils. The migration of neutrophils to Aβ could impair cognitive function [36,37]. Additionally, a crucial component of treating AD is inhibiting neutrophil elastase and neutrophil transport [38,39,40]. Overactivation of neutrophils is an essential feature of AD [41]. Secondly, we realized a biological pathway associated with platelets, which are crucial to the pathophysiology of AD. Platelets create Aβ and other amyloid precursor protein (APP) secretase products [42]. Thrombocyte degranulation releases APP (sAPP) secreted by recombination, which will lead to negative feedback regulation during platelet activation [43]. Compared with the control group, AD patients retain more APP in their activated platelets [44]. The pathophysiology of AD may also be impacted by serotonin [45]. Other biological mechanisms that are implicated to AD include phagocytosis and the response to hydroperoxide [46,47,48].

### 4.4. Regression Analysis

According to Table 4, the gray matter volume of the brain was regressed using three conventional regression techniques and their R2 scores were all fairly close to 1. The result demonstrated that all three regression approaches had outstanding model fit and strong regression performance. It also clearly indicated how closely the amount of gray matter in the brain correlates with the genes appointed by the proposed algorithm. We confirmed the validity of the approach in this work for detecting genetic connections with QTs of MRI-derived ROI measurements in the ADNI Cohort.

The contour coefficient of the subspace clustering algorithm through a multilayer neural network was larger than that of the subspace clustering algorithm without a multilayer neural network in the gene, T2, T3, and T4 periods (Table 7). Additionally, in the T1 period, the contour coefficients of the two were similar. The mean ± standard deviation of five indices, such as F-score, which had passed through the subspace clustering algorithm of the multilayer neural network 20 times with different initialization, was basically larger than that of the subspace clustering algorithm without the multilayer neural network in the gene, T1, T2, T3 and T4 periods. It was obvious that the multilayer neural network enabled the subspace clustering technique to achieve superior clustering performance.

Figure 7 illustrated how little disturbance was present in the heatmap created using the reconstructed data, regardless of whether it represented brain regions or gene expression data, and how this results in a more pronounced pattern, demonstrating how the features selected by the reconstructed data were more stable and could be chosen more precisely. The distinguishing characteristics attested to the efficiency of the deep subspace reconstruction algorithm.

## 5. Conclusions

We proposed a combined method of deep subspace reconstruction and hypergraph-based TGSCCA (DS-HBTGSCCA), which investigated the nonlinear features and high-order correlation between longitudinal sMRI data and gene expression data. It was successfully applied to the identification of AD biomarkers. We compared the performance of TGSCCA and DS-HBTGSCCA on simulated data sets and real data sets, respectively. The experimental results showed that the biological significance analysis of the top 10 brain regions and top 20 genes discovered by DS-HBTGSCCA confirmed that this method could accurately detect the AD-related pathogenic ROIs and genes, and that two risk brain regions (left middle temporal and right inferior temporal) with significant differences in time sequence were found in the top 10 brain regions. In addition, a regression analysis of the top brain areas was carried out, and the effectiveness of the deep subspace strategy was verified.

However, there are some limitations in this study. There was a small number of AD subjects compared to other groups. An unbalanced number of different types of samples often leads to deviations in results. In future research, we aim to use SMOTE and other resampling methods and collect more samples to overcome this problem. In addition, more genetic and clinical data will be taken into account in imaging genetics research to explore complex relationships between the data. Meanwhile, developing deep learning techniques for studying the correlation between longitudinal images and genetic data and recognizing nonlinear changes in features over time would be an interesting research direction.

## Figures and Tables

**Figure 1 biomolecules-13-00728-f001:**
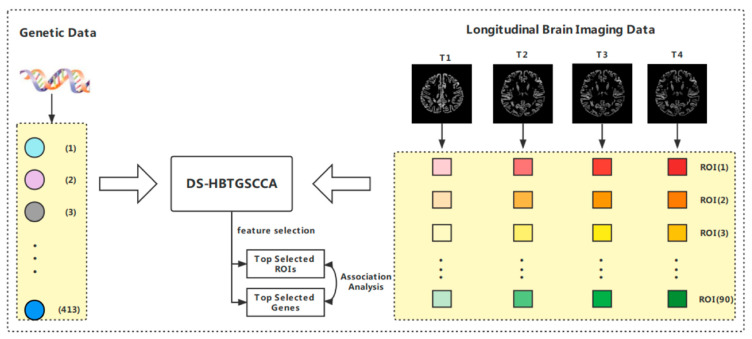
The schematic diagram of the proposed algorithm DS-HBTGSCCA.

**Figure 2 biomolecules-13-00728-f002:**
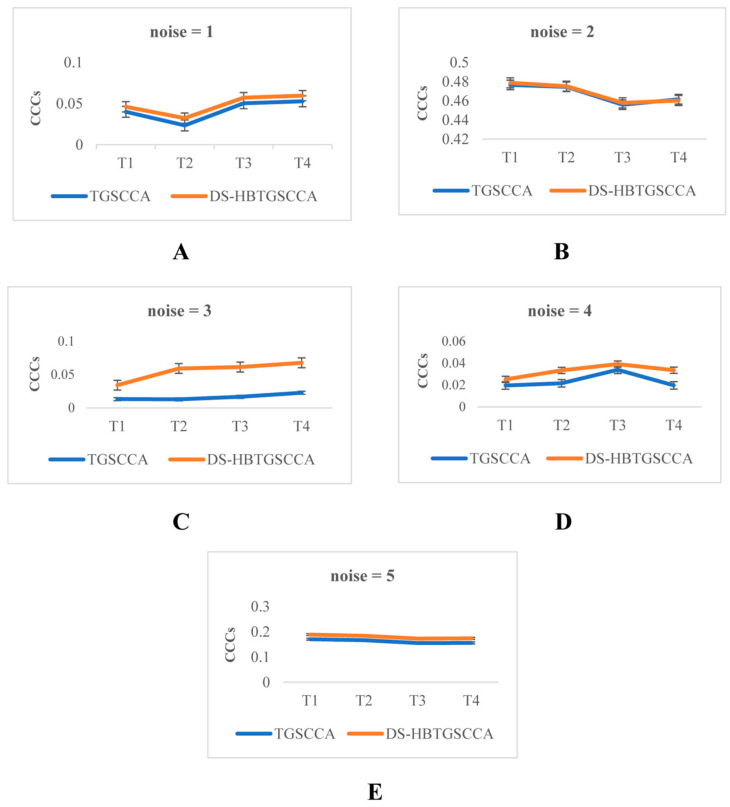
Comparison of *CCCs.* (**A**–**E**) represent the *CCC* line graphs of the two methods in four different periods when the noise level is from 1 to 5, respectively.

**Figure 3 biomolecules-13-00728-f003:**
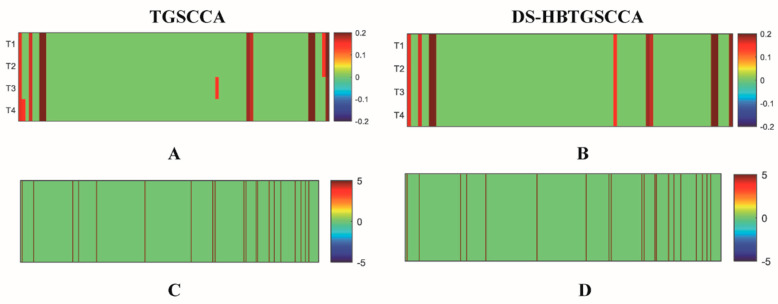
The estimating weight of the top 10 ROIs and the top 20 genes on reconstructed data using two methods. (**A**,**B**) are the heatmaps of the top 10 ROIs selected by two algorithms using the reconstructed data, and T1–T4 represent four periods; (**C**,**D**) are the heatmaps corresponding to the top 20 genes selected by two algorithms using the reconstructed data.

**Figure 4 biomolecules-13-00728-f004:**
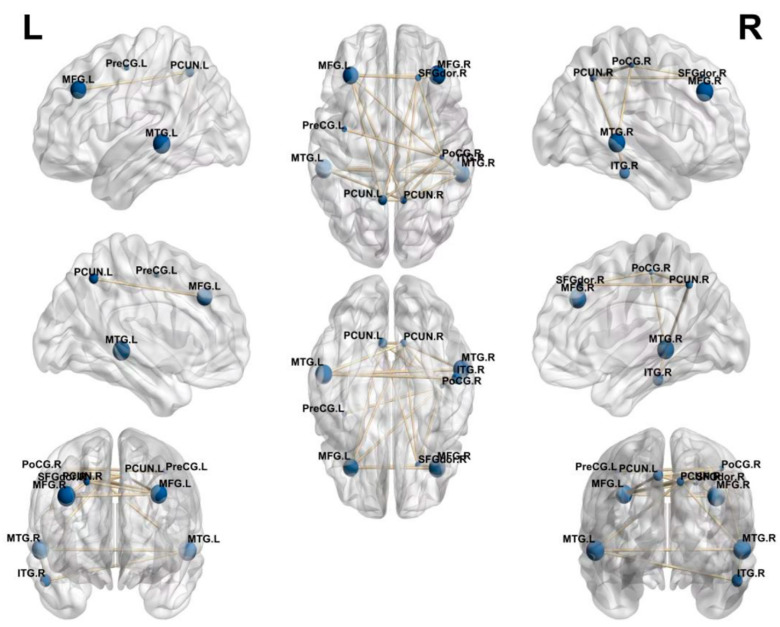
Visualization of the top 10 ROIs selected by the DS-HBTGSCCA.

**Figure 5 biomolecules-13-00728-f005:**
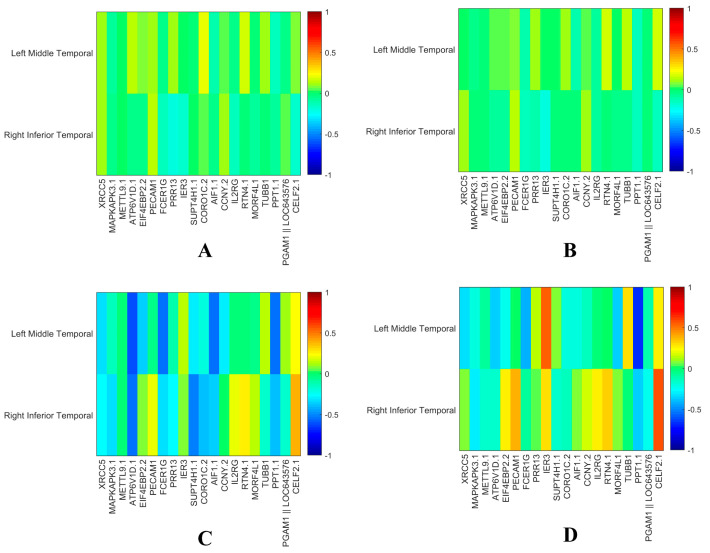
ROI−gene correlation heatmaps. (**A**,**B**) represent ROI−gene correlation heatmaps of HC at T1 and T4, respectively; (**C**,**D**) represent ROI−gene correlation heatmaps of AD in T1 and T4, respectively.

**Figure 6 biomolecules-13-00728-f006:**
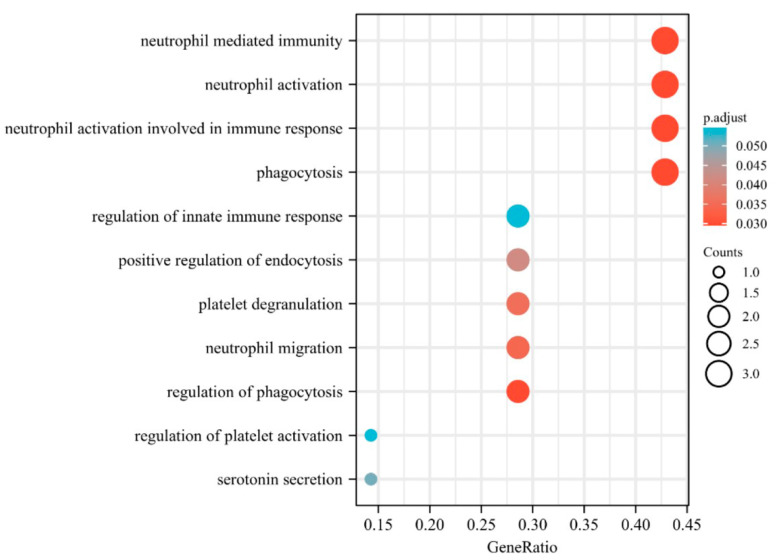
GO enrichment analysis results of the top 20 genes.

**Figure 7 biomolecules-13-00728-f007:**
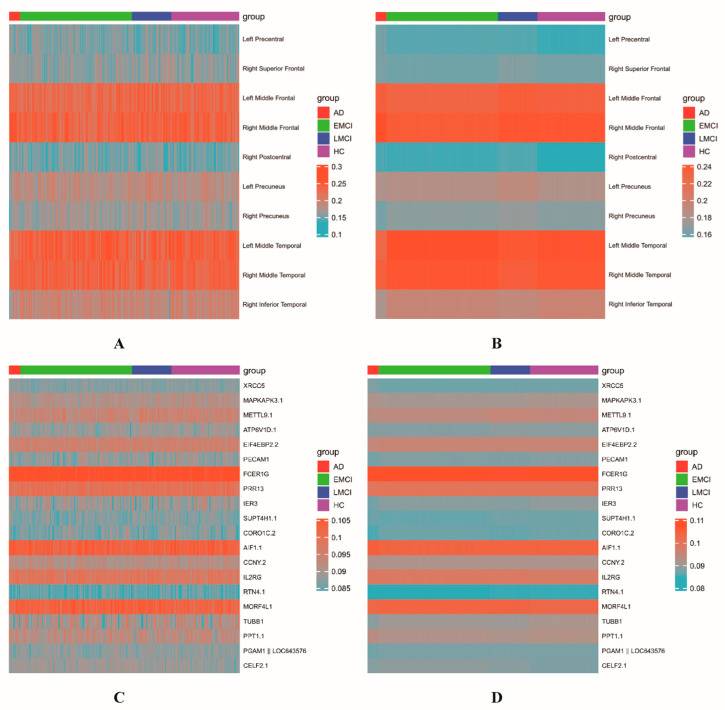
Visual display of two kinds of data. (**A**,**C**) are the heatmaps corresponding to the top 10 ROIs and top 20 genes of the original data, respectively; (**B**,**D**) are the heatmaps corresponding to the top 10 ROIs and top 20 genes of the reconstructed data, respectively.

**Table 1 biomolecules-13-00728-t001:** The demographics of the participants.

Groups	AD	EMCI	LMCI	HC
Number	10	102	36	62
Gender(M/F)	6/4	55/47	18/18	34/28
Age (mean ± std)	79.65 ± 9.77	70.95 ± 7.07	73.40 ± 7.19	75.83 ± 5.83

**Table 2 biomolecules-13-00728-t002:** *CCCs* on the reconstructed ADNI data.

Algorithm	T1	T2	T3	T4
TGSCCA	0.6881	0.6754	0.6905	0.6799
DS-HBTGSCCA	0.7169	0.6967	0.6884	0.665

**Table 3 biomolecules-13-00728-t003:** *CCCs* on the original ADNI data.

Algorithm	T1	T2	T3	T4
TGSCCA	0.1037	0.1138	0.1509	0.1699
DS-HBTGSCCA	0.2186	0.185	0.2036	0.1271

**Table 4 biomolecules-13-00728-t004:** The top 10 ROIs and canonical weights (absolute values) in two algorithms.

	T1		T2		T3		T4	
ROI	TGSCCA	DS-HBTGSCCA	TGSCCA	DS-HBTGSCCA	TGSCCA	DS-HBTGSCCA	TGSCCA	DS-HBTGSCCA
Left Middle Temporal	0.242891	0.241756	0.24247	0.241756	0.241243	0.241756	0.240415	0.241756
Right Middle Frontal	0.239456	0.239911	0.239386	0.239911	0.240085	0.239911	0.240716	0.239911
Right Middle Temporal	0.239204	0.238587	0.239034	0.238587	0.238354	0.238587	0.237752	0.238587
Left Middle Frontal	0.231468	0.231956	0.231756	0.231956	0.23217	0.231956	0.232429	0.231956
Right Inferior Temporal	0.19458	0.193725	0.19405	0.193725	0.1936	0.193725	0.192668	0.193725
Left Precuneus	0.185073	0.185278	0.185006	0.185278	0.185381	0.185278	0.185652	0.185278
Right Precuneus	0.170217	0.170441	0.170298	0.170441	0.170519	0.170441	0.17073	0.170441
Right Superior Frontal	0.163554	0.163951	0.163448	0.163951	0.163973	0.163951	0.164824	0.163951
Left Precentral	0.156793	0.156955	0.156787	0.156955	0.157135	0.156955	0.157101	0.156955
Right Postcentral	0.154319	0.154576	0.154386	0.154576	0.154746	0.154576	0.154849	0.154576

**Table 5 biomolecules-13-00728-t005:** The top 20 genes and canonical weights in DS-HBTGSCCA (absolute values).

Gene	Weight
FCER1G	0.02043
AIF1.1	0.018494
MORF4L1	0.018055
PRR13	0.016633
IL2RG	0.015529
EIF4EBP2.2	0.014426
METTL9.1	0.014098
PPT1.1	0.013179
CCNY.2	0.013001
TUBB1	0.012844
MAPKAPK3.1	0.012789
IER3	0.01205
ATP6V1D.1	0.011843
PECAM1	0.011714
CELF2.1	0.011599
PGAM1 || LOC643576	0.011451
CORO1C.2	0.011208
XRCC5	0.011188
SUPT4H1.1	0.011082
RTN4.1	0.010388

**Table 6 biomolecules-13-00728-t006:** The mean *R^2^*_score of the top 10 ROIs obtained by three regression methods.

ROI	Regression Method (Mean of *R^2^*_Score)		
	Bayes	LR	RD
Left Middle Temporal	0.953981	0.952863	0.801004
Right Middle Frontal	0.964647	0.958669	0.905144
Right Middle Temporal	0.970721	0.962353	0.956769
Left Middle Frontal	0.96607	0.945351	0.965932
Right Inferior Temporal	0.964978	0.932218	0.967617
Left Precuneus	0.972586	0.941337	0.974177
Right Precuneus	0.971747	0.940244	0.974972
Right Superior Frontal	0.967789	0.936142	0.977111
Left Precentral	0.968347	0.92251	0.977113
Right Postcentral	0.966746	0.907586	0.977428

**Table 7 biomolecules-13-00728-t007:** Contour coefficients of subspace clustering algorithm through multilayer neural network and subspace clustering algorithm without multilayer neural network.

		Contour Coefficient	F-Score	Precision	Recall	NMI	Adj-RI
Through multilayer neural network	Gene	0.7675	0.4599 ± 0.0358	0.5267 ± 0.0486	0.4083 ± 0.0286	0.2747 ± 0.0514	0.2198 ± 0.0556
	T1	0.4667	0.3154 ± 0.0150	0.3610 ± 0.0045	0.2810 ± 0.0259	0.0259 ± 0.0043	0.0117 ± 0.0062
	T2	0.5052	0.3655 ± 0.0289	0.3983 ± 0.0107	0.3391 ± 0.0417	0.0374 ± 0.0032	0.0642 ± 0.0190
	T3	0.4723	0.2941 ± 0.0050	0.3532 ± 0.0035	0.2519 ± 0.0059	0.0198 ± 0.0060	0.0019 ± 0.0042
	T4	0.4732	0.3462 ± 0.0088	0.3403 ± 0.0040	0.3529 ± 0.0190	0.0377 ± 0.0107	0.0181 ± 0.0070
Without multilayer neural network	Gene	0.2478	0.3615 ± 0.0104	0.3709 ± 0.0056	0.3535 ± 0.0107	0.0786 ± 0.0073	0.0282 ± 0.0059
	T1	0.4731	0.3130 ± 0.0117	0.3552 ± 0.0037	0.2801 ± 0.0171	0.0142 ± 0.0060	0.0046 ± 0.0048
	T2	0.4544	0.3087 ± 0.0061	0.3416 ± 0.0016	0.2817 ± 0.0097	0.0225 ± 0.0048	0.0133 ± 0.0022
	T3	0.4414	0.3233 ± 0.0138	0.3594 ± 0.0077	0.2977 ± 0.0485	0.0354 ± 0.0106	0.0089 ± 0.0114
	T4	0.4242	0.3022 ± 0.0068	0.3536 ± 0.0047	0.2641 ± 0.0098	0.0127 ± 0.0062	0.0023 ± 0.0056

Note: The data above were represented as mean ± standard deviation (SD) of subspace clustering algorithm with different initialization after 20 times.

## Data Availability

The authors appreciate the Alzheimer’s Disease Neuroimaging Initiative (ADNI) for contributing data (https://adni.loni.usc.edu (accessed on 15 March 2022)).

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
