# Peer review of "A Novel Longitudinal Phenotype–Genotype Association Study Based on Deep Feature Extraction and Hypergraph Models for Alzheimer’s Disease"

_biomolecules, 2023, doi:10.3390/biom13050728_

Round 1

Reviewer 1 Report

In this manuscript, authors proposed a novel method that combines deep subspace clustering with hypergraph-based temporally-constrained group Sparse Canonical Correlation Analysis (DS-HBTGSCCA) to discover the deep association between longitudinal phenotypes and genotypes.

Authors presented some data in the results section and the remaining data in the discussion section. I suggest merging both sections together, this will make the presentation of data clearer and more appropriate than two separate sections.

Authors need to clarify and explain some of their findings as indicated in my comments.

Methods

2.1. Data and preprocessing: the title is not accurate, it could be “Study population”.

Line 90: Please mention the full name “Alzheimer’s Disease Neuroimaging Disease” and put the abbreviation (ADNI) between brackets.

Line 91: Please mention the full names for EMCI, LMCI and HC in the text (not only in the table footnote) and put the abbreviation of each between brackets.

Small number of AD subjects compared to other groups, can you mention the reason in the study? And adding in the conclusion section a sentence about the necessity of confirming the study findings in a larger number of AD patients. 

Table 1 represents the demographics of study population, I suggest restating the title.

Line 102: Please add the full name; regions of interest (ROIs).

Line 108: Please correct P>0.05 instead of P<0.05.

Please add a section of “Statistical analysis” at the end of Methods section indicating the statistical tests and significance level used in the study, and the software used to generate the graphs and images in the study.

Results

3.1. On simulation data

I suggest adding Section 192 1.3 of the supplementary materials in the main text not the supplementary materials.

In T1, T2, and T3 periods, our algorithm is better than TGSCCA. Please explain this result, I don’t see any difference between your algorithm and TGSCCA in Figure 2B.

Figure 3: Please label the orange set data as “DS-HBTGSCCA” not Ours. What is the significance in T1 and T2 comparisons? Please indicate that on the figure. Please add the word data after ADNI in the figure title.

Discussion

Authors stated at the end of results section that “By identifying these risk genes and brain risk regions, we can better understand how AD develops and establish the foundation for its future medical treatment”, then they explained that at the beginning of the discussion section. Please merge both results and discussion sections. Tables S1 and S2 must be in the main manuscript not the supplementary materials. These tables represent main part of the study findings.

Line 270: Right Middle Temporal (P = 0.055), this p-value is close to significance but not significant, please correct that.

Line 300: Please add the full name; thrombospondin-1 (TSP-1).

Which bioinformatic tool was used for the GO enrichment analysis? Please provide more details about the analysis parameters.

Table 3 title could beContour coefficients of subspace clustering algorithm through multilayer neural network and subspace clustering algorithm without multilayer neural network”. Below the table, you can add data are represented as mean ± standard deviation (SD) of subspace clustering algorithm with different initialization after 20 times.

Manuscript requires moderate English editing.

Reviewer 2 Report

The whole article is well organized. However, several questions should be asked.

First, as mentioned in the introduction, the research group proposed a novel algorithm named DS-HBTGSCCA that combines deep subspace clustering and hypergraph- based Time Constraint Group SCCA (TGSCCA) algorithm provided a creative combination of two neural networks. However, the reason for choosing deep subspace clustering to handle the sMRI data was put into the conclusion part, which should be explained in the introduction.

2. It Is known that deep subspace clustering belongs to the deep auto-encoders, an unsupervised machine learning algorithm. The connection between the sMRI image and the patient’s data also depends on the number of patients. Compared to the other three groups, AD patients only account for n=10. What is the subspace clustering error in DS-HBTGSCCA (mainly on the deep subspace clustering part)? Does DS- HBTGSCCA still perform better than TGSCCA when the n number increases?

3. Interestingly, the time-constraint group usually participates in the research on the progress of disease or the effect of aging in AD; the use of the time-constraint group in determining gene targets was seldom raised in the topic. So, what are the differences between the gene targets identified through DS-HBTGSCCA and thetraditional method-identified gene targets in AD?

4. Apart from the differences, there are similar research proposed an algorithm to analyze single-cell sequencing data combined with MRI images, which shows a strong connection between different types of cells and the correlated gene expression in each part of the brain in the post-mortem brain images, the proposed DS- HBTGSCCA shows much weaker connections between sMRI image and the patient’s data, would it be a better way to conclude the effect of changes of this connections with respect to time?

5. As found from the results of GO analysis, the identified gene targets almost all belong to AD-correlated gene groups. And so, compared to a traditional bioinformatic analysis like GO or KEGG analysis of differentially expressed genes, what are the pros and cons of using DS-HBTGSCCA? Furthermore, time spent, computer resources, accuracy, and other possible parameters should also be discussed.

Round 2

Reviewer 1 Report

Authors addressed most of my comments in the first revision, but I still see that the study data are divided between the results section and discussion section. All the data should be included under the results section and discussion of study data and comparing them to literature should be under the discussion section.

Author Response

Thank you for your valuable comments and suggestions on this manuscript. I have adjusted the the study data of the article to the result part, and the discussion of study data and comparison with literature to the discussion part. And the details of the full text have been carefully revised again.

Reviewer 2 Report

questions are answered, accepted. 

Author Response

Thank you for your valuable comments on this manuscript.